# Solventless Photopolymerizable Paper Coating Formulation for Packaging Applications

**DOI:** 10.3390/polym15051069

**Published:** 2023-02-21

**Authors:** Fábio M. Silva, Ricardo J. B. Pinto, Ana Barros-Timmons, Carmen S. R. Freire

**Affiliations:** CICECO—Aveiro Institute of Materials, Chemistry Department, University of Aveiro, 3810-193 Aveiro, Portugal

**Keywords:** paper-based materials, photopolymerizable coatings, solvent-free formulations, hydrophobicity, packaging materials

## Abstract

Nowadays, packaging applications require the use of advanced materials as well as production methods that have a low environmental impact. In this study, a solvent-free photopolymerizable paper coating was developed using two acrylic monomers (2-ethylhexyl acrylate and isobornyl methacrylate). A copolymer, with a molar ratio of 2-ethylhexyl acrylate/isobornyl methacrylate of 0.64/0.36, was prepared and used as the main component of the coating formulations (50 and 60 wt%). A mixture of the monomers with the same proportion was used as a reactive solvent, yielding formulations with 100% solids. The coated papers showed an increase in the pick-up values from 6.7 to 32 g/m^2^ depending on the formulation used and the number of coating layers (up to two). The coated papers maintained their mechanical properties and presented improved air barrier properties (Gurley’s air resistivity of ≈25 s for the higher pick-up values). All the formulations promoted a significant increase in the paper’s water contact angle (all higher than 120 °) and a remarkable decrease in their water absorption (Cobb values decrease from 108 to 11 g/m^2^). The results confirm the potential of these solventless formulations for fabricating hydrophobic papers with potential application in packaging, following a quick, effective, and more sustainable approach.

## 1. Introduction

Packaging is one of humanity’s most important commodities, responsible for ensuring the stability and safety of its contents [1]. However, it is also becoming a major environmental concern, not only due to the materials currently used (mostly plastics coming from non-renewable sources) but also due to the large amounts of waste it produces [2]. To solve this problem, innovative solutions involving new materials and new production methods capable of providing more sustainable packaging solutions are in demand [3,4]. Paper is a suitable material to be used for sustainable packaging since it is mainly composed of cellulose fibers, and therefore it is biodegradable and a lightweight material with good mechanical properties [5,6]. However, the hydrophilicity and porosity of this cellulosic substrate limit its use as a packaging material [7,8]. Thus, the application of a coating appears as one interesting alternative to improve the barrier properties of paper, as well as to impart other functionalities [5,9,10].

Most of the materials used as coatings to fabricate papers with good barrier properties are synthetic polymers obtained from non-renewable sources, such as polyolefins, poly(ethylene-vinyl alcohol) (PEVOH), and poly(vinylidene chloride) (PVDC) [6,11]. Apart from being non-renewable materials, they are usually applied using volatile organic compounds (VOCs) as solvents, posing an environmental problem in terms of air pollution, not to mention the health hazard they represent to whoever handles them [12,13]. Even more sustainable alternatives, such as the bio-sourced poly(lactic acid) (PLA), a compostable polymer with inherent hydrophobicity [14,15], presents some major drawbacks regarding its processability and application as coating material since it also requires, in most cases, the use of volatile organic solvents [8,10], and presents a melt instability above its melting temperature, having a narrow window of processability when extruded [6]. Therefore, the implementation of photopolymerizable coatings can be a major advantage in the quest to develop more sustainable coating systems [16]. Photopolymerizable coating formulations are mostly composed of oligomers/polymers, a reactive solvent, and a photoinitiator. The oligomer/polymer is usually the major constituent of the formulation and determines the main properties of the coating [17]. The reactive solvent serves as a rheology controller and is generally constituted by one or more monomers that will polymerize after the application of the coating formulation and its subsequent irradiation. This is the most important feature of photopolymerizable coatings, because by using a reactive solvent, the coating is considered to have 100% of solids, i.e., all the deposited formulation becomes part of the final coating layer, and the use of organic solvents or water is avoided, contributing to the reduction of VOCs, wastewater, and energy. Also, the cure can be carried out in a matter of seconds by the incidence of ultraviolet (UV) light [18]. Despite the obvious advantages of photopolymerizable coatings, there are very few studies regarding their application to paper, especially to obtain paper-based products for packaging.

The monomer 2-ethylhexyl acrylate has been widely studied as a component of photopolymerizable formulations, mostly for the development of pressure-sensitive adhesives [19,20,21,22,23,24]. However, due to the low (−52 °C) glass transition temperature (*T_g_*) of the corresponding homopolymer, it is usually combined with other monomers, such as methyl methacrylate, to yield polymers with higher *T_g_* [25]. Isobornyl methacrylate is a good alternative as a co-monomer since its homopolymer presents a *T_g_* of around 190 °C, and it is derived from pine resin, being thus a bio-sourced monomer [26]. To the best of our knowledge, the combination of these two monomers has only been reported once for the development of pressure-sensitive adhesives [27]. However, despite the fact that the hydrophobicity of the homopolymers of these two monomers is well known [28,29,30,31], their use to prepare hydrophobic photopolymerizable coatings, and their application as paper coatings, have not been previously reported. Thus, in this study, we describe the preparation of a more sustainable photopolymerizable coating composed of these two acrylic monomers, capable of imparting paper with high hydrophobicity and water barrier properties. Furthermore, other important properties for packaging applications, such as mechanical and air barrier properties, maintained or even improved.

## 2. Materials and Methods

### 2.1. Materials

The monomers isobornyl methacrylate (technical grade) and 2-ethylhexyl acrylate (98%), the photoinitiator (PI) 2-hydroxy-2-methylpropiophenone (97%), and the thermal initiator 2,2′-azobis(2-methylpropionitrile) (AIBN, 98%) were provided by Sigma-Aldrich. Chloroform (99.5%), methanol (99.8%), and tetrahydrofuran (THF, 99.8%) were supplied by Fisher Chemical. 1-Dodecanethiol (98%), provided by Sigma-Aldrich, was used as a chain transfer agent. All commercial products were used as received. Base paper (Navigator, 70 g/m^2^) was provided by The Navigator Company (Portugal). Ultrapure water (Type 1, 18.2 MΩ·cm at 25 °C) was obtained via a Simplicity Water purification system (Merck, Darmstadt, Germany).

### 2.2. Synthesis of Poly(2-Ethylhexyl Acrylate-Co-Isobornyl Methacrylate)

The synthesis of the copolymer was adapted from a procedure described in the literature [19]. In a two-neck, round-bottom flask, 2-ethylhexyl acrylate and isobornyl methacrylate were added in a molar ratio of 0.64/0.36, respectively, aiming to yield a copolymer with a *T_g_* of 0 °C. The ratio of monomers used to produce a copolymer with this *T_g_* was determined using the Fox equation (Equation (1)), which relates the final glass transition temperature (*T_g,mix_*) of the copolymer with the ratio of co-monomers weight fractions (*w*_1_ and *w*_2_) and the *T_g_* of the respective homopolymers (*T_g_*_,1_ and *T_g_*_,2_) [32].
(1)$$\frac{1}{T_{g,mix}}=\frac{w_{1}}{T_{g,1}}+\frac{w_{2}}{T_{g,2}}$$

0.5 wt% of AIBN and 0.7 wt% of 1-dodecanethiol (both weight percentages relative to the mass of monomers used) were added to the mixture of monomers. The reaction mixture was purged at ambient temperature, by bubbling N_2_ for 30 min under stirring at 600 rpm, and then the temperature was raised to 80 °C. After reaching the required temperature, the mixture was allowed to react for 30 min more (while in N_2_ atmosphere) and finally left to cool to room temperature exposed to air. The obtained copolymer was purified via consecutive steps of solubilization in chloroform followed by precipitation in cold methanol; the process was repeated three times. Finally, the copolymer was dissolved in chloroform and dried using a rotary evaporator.

### 2.3. Characterization of the Copolymer

Size exclusion chromatography (SEC) analysis of the synthesized copolymer was performed using a PL-GPC 220 system (Agilent, Santa Clara County, CA, USA) equipped with two Agilent PL gel MIXED-D columns, 7.5 × 300 mm, 5 µm (in series) and protected by a PL gel MIXED precolumn, 7.5 × 50 mm, 5 µm. The columns, injector system, and detector (RI) were maintained at 40 °C during the analysis. The eluent used was THF, and it was pumped at a flow rate of 0.5 mL min^−1^. The columns were calibrated using a polystyrene solution (370–187,700 g mol^−1^), a standard polymer supplied with the equipment. Four copolymer samples from different batches were analyzed to determine the molecular weight (Mw), and the result is presented as a mean value.

^1^H nuclear magnetic resonance spectroscopy (^1^H NMR) analyses of the copolymer samples dissolved in CDCl_3_-d and using tetramethylsilane (TMS) as an internal standard, were carried out using a Bruker AMX 300 Spectrometer operating at 300.13 MHz. The ^1^H NMR spectrum of the copolymer is presented in Appendix A and was used to determine the copolymer composition by identifying the chemical shifts associated exclusively to each of the repeating units of the copolymer. In this case, the chemical shifts at 4.40 and 3.84 ppm were assigned to the hydrogen of the *ortho*-carbon of the isobornyl methacrylate’s ring, and the two hydrogens of the methylene group linked to the oxygen of the ester group of 2-ethylhexyl acrylate, respectively (marked as 1 and 2 in the copolymer structure shown in Figure 1) [25]. Then, the percentage of each repeating unit was calculated using Equations (2) and (3):(2)$$x_{2-ethylhexyl\;acrylate}=\frac{\frac{A_{3.84}}{2}}{\frac{A_{3.84}}{2}+A_{4.40}}$$
(3)$$x_{isobornyl\;methacrylate}=1-x_{2-ethylhexyl\;acrylate}$$
where *A*_3.84_ and *A*_4.40_ refer to the area of the peaks at 3.84 and 4.40 ppm, respectively, and *x_isobornyl methacrylate_* and *x*_2-*ethylhexyl methacrylate*_ refer to the molar ratio of isobornyl methacrylate and 2-ethylhexyl acrylate, respectively.

Dynamic mechanical analysis (DMA) of the copolymer was carried out to determine its *T_g_*. This analysis was carried out using a Tritec 2000 DMA instrument (Triton Technology, Leicestershire, UK) and a stainless-steel material pocket accessory, in single cantilever bending mode. The temperature scan was carried out from −100 °C up to 100 °C at a heating rate of 5 °C·min^−1^, with a displacement of 0.020 mm, at two frequencies of deformation (oscillating frequency): 1 and 10 Hz.

### 2.4. Preparation of Coating Formulations

The coating formulations were prepared by mixing the copolymer, the reactive solvent consisting of the two monomers at the same ratio as used for the synthesis of the copolymer, and 5 wt% of PI (weight percentage in relation to the sum of the mass of copolymer and reactive solvent used), under magnetic stirring until homogeneous solutions were obtained. Two formulations were prepared, one containing 50 and another 60 wt% (weight percentage in relation to the total mass of copolymer and reactive solvent used) of the copolymer.

### 2.5. Coating of Paper Samples

All coated paper samples (4 cm × 4 cm) were prepared using a manual coater (K Lox Proofer, RK Print coat instruments). After the coating application, the samples were immediately irradiated with UV light for 300 s (in air) using a BlueWave^®^ 200 version 3.0, from Dymax, with a radiance of 150 mW/cm^2^. All coated papers were conditioned in a room with controlled temperature (23 ± 1 °C) and humidity (50 ± 1%), for at least 12 h, following the standard ISO 187 before characterization.

### 2.6. Characterization of the Coated Papers

The optical properties of the coated papers (whiteness, brightness, and opacity) were characterized following the standards ISO 2470-2 for whiteness, ISO 2470-1 for brightness, and ISO 2471 for opacity, using a spectrophotometer L&W ELREPHO (ABBKista, Seden, Sweden). For the measurement of whiteness and brightness, 10 samples were analyzed while stacked. For the opacity measurement, the same stack of 10 samples was used, but for each sample individual sheets were also measured.

The Fourier transform infrared-attenuated total reflection (FTIR-ATR) spectra of all samples were collected using a Perkin-Elmer FT–IR System Spectrum BX spectrophotometer (Perkin-Elmer Inc., Waltham, MA, USA) equipped with a single horizontal Golden Gate ATR cell, using 32 scans with a spectral resolution of 4 cm^−1^ in a scanning range from 500 to 4000 cm^−1^. The base paper (without coating) was used as the background.

Scanning electronic microscopy (SEM) analysis was performed using a FE-SEM Hitachi SU-70 (Hitachi High-Technologies Corporation, Tokyo, Japan) operated at 15 kV. All samples were coated with a carbon film before the analysis.

The air barrier properties of the coated samples were evaluated using the Gurley test, following the ISO 5636-5 standard. The device used for the test was a L&W 121E tester (ABB, Portugal), which measures the time that was required for 100 mL of air to flow through the sample. For this test, ten paper samples were analyzed.

The burst strength of the papers was measured using a device that applies negative pressure onto the paper until failure, performed on a burst tester (model S185340000, Frank-PTI, Birkenau, Germany). Likewise, in this analysis, ten samples were measured for each condition studied.

The water contact angles of the coated papers were measured using a Dataphysics Contact Angle System OCA-20 (Norleq, Porto, Portugal) and ultrapure water as the liquid. For this analysis, ten measurements were made for each paper sample at room temperature, using droplets of 5 µL of water.

The Cobb test was carried out following the ISO 535 standard. In this case, five paper samples were evaluated for each condition studied. Each sample was put in contact with a column of water (with 1 cm of height) for 60 s. The tested area was 10 cm^2^.

All results are presented as a mean, with its corresponding standard deviation (SD). Statistical analyses were performed for all the measurements, and the results were compared using the one-way analysis of variance (ANOVA) complemented with Tukey’s tests, using Origin 9 (OriginLab Corporation, Northampton, MA, USA).

## 3. Results and Discussion

The present work consisted of the development of new photopolymerizable coating formulations capable of imparting paper with high water barrier properties, using two acrylic monomers, including one biobased monomer, viz, isobornyl methacrylate, according to the methodology illustrated in Figure 1.

Poly(2-ethylhexyl acrylate-*co*-isobornyl methacrylate) with a monomer ratio of 0.6/0.4, as determined by ^1^H NMR (Appendix A), was prepared via conventional free radical polymerization in the presence of 1-dodecanethiol to promote chain transfer reactions, thus preventing the formation of copolymers with very high molecular weights [19]. Considering that it was used a small amount of the chain transfer, the Mw of the ensuing copolymer was still 19699 g/mol (polydispersity index, *Ð* = 1.76). In turn, the *T_g_* of the copolymer was −1 °C, which is remarkably close to that estimated by the Fox law equation. In fact, the small discrepancy (just 1 °C) might be justified by the presence of the dodecyl end-groups associated with the use of the chain transfer agent.

Next, the coating formulations were prepared by mixing the copolymer and the reactive solvent (composed of the same monomers in the same molar ratio used to synthesize the copolymer). Two formulations with different mass ratios of the copolymer were studied, namely 50 and 60 wt%. These amounts of copolymer were selected based on a preliminary test that allowed us to verify that formulations with copolymer mass ratios higher than 60 wt% could not be applied to paper by roll-to-roll methods due to their very high viscosity. Thus, the mass ratios of 50 and 60 wt% were chosen to keep the amount of reactive solvent used as low as possible while maintaining the processability of the formulations. Finally, both coating formulations were applied using a manual coater and photopolymerized, yielding different coated paper samples. To study the impact of both formulations on the properties of paper, one and two coating layers were applied onto 4 cm × 4 cm paper sheets. Table 1 summarizes the grammage and pick-up values of the obtained coated papers. From the comparison of the pick-up results of these coated paper samples, it is clear that the P60% formulation affords higher pick-up values. This fact is explained by the higher copolymer content in this formulation and, consequently, a higher viscosity that results in the deposition of an increased amount of formulation. Furthermore, as expected, a significant increase in the pick-up was observed when two layers of the formulation were deposited. Specifically, for the P50% formulation, the pick-up value (P50%_2) increased 1.55 times. This increase was even higher for the P60% formulation, which more than doubled (P60%_2).

FTIR-ATR spectra of the coated papers collected right after the coating application and after UV irradiation (Figure 2) were compared to assess the success of the photopolymerization. The non-irradiated samples presented a small band at 1636 cm^−1^ corresponding to the stretching vibration of the C=C bond of the monomers (2-ethylhexyl acrylate and isobornyl methacrylate) [33,34], while the cured samples do not show this band. This is a confirmation that the reactive solvent present in the cured samples was totally polymerized. Other characteristic vibrations of the polymeric coating include the peaks at 2956 cm^−1^, 2930 cm^−1^, and 2874 cm^−1^, attributed to the stretching vibrations of the C–H bonds, and the peak at 1726 cm^−1^ attributed to the stretching vibration of the C=O of the ester groups, being all of them present both in the copolymer and the monomers [33,34].

### 3.1. Morphology

SEM analysis was carried out to assess the surface morphology of the coated papers, prepared with both formulations and different number of coating layers. The micrographs of the coated papers and the pristine base paper, used for comparative purposes, are presented in Figure 3. The uncoated paper (Figure 3a) shows the randomly oriented cellulose fibers as well as fillers (mostly precipitated calcium carbonate) frequently used in paper manufacture [35]. In the micrographs of the coated samples (Figure 3b–e), the presence of the coating is essentially confirmed by the partial coverage of the fillers present on paper. Moreover, increasing the number of coating layers (and the final coating pick-up) resulted in a more evident coating on top of the paper, a feature also reported for other types of coatings based, for example, on chitosan [36], or poly(3-hydroxybutyrate) (PHB) [11].

### 3.2. Optical Properties

The optical properties of paper, specifically whiteness, brightness, and opacity, have a major impact on paper’s printability and color reproducibility [37,38]. The effect of the different formulations and the number of coating layers on these properties were evaluated by colorimetry (Figure 4). The application of these two formulations had the same impact on the whiteness reflectance factor of the papers (Figure 4a). The application of one layer reduced the whiteness reflectance from 137.83 ± 0.81% (base paper) to 121.18 ± 0.69 and 120.05 ± 1.45% for the formulations P50% and P60%, respectively. The deposition of a second layer further reduced the whiteness reflectance for both formulations (116.04 ± 1.31% for P50% and 116.43 ± 1.15% for P60%).

The brightness of the coated papers (Figure 4b) follows a similar trend to that observed for whiteness. A single layer of both formulations led to a similar reduction in the reflectance factor values of the base paper, decreasing from 104.01 ± 0.03% to 95.23 ± 0.42 and 92.77 ± 1.03% for P50% and P60%, respectively. The application of a second layer of the formulations also had a significant impact on the brightness’ reflectance values. In fact, a reduction to 92.23 ± 0.45% for P50% and to 86.65 ± 1.37% for P60% was registered, suggesting that this property is also influenced by the pick-up increase. The high values of whiteness and brightness of paper are mainly associated with the use of fillers (such as precipitated calcium carbonate), which scatter the visible light [39]. With the application of an organic coating (such as in the present study), the light scattering provided by the fillers is dampened, resulting in the reduction of the whiteness and brightness observed [40].

As regards the opacity of the paper samples (Figure 4c), the base paper presented an opacity of 93.85 ± 0.1%, decreasing to 92.27 ± 1.15% and 87.76 ± 3.18% with the application of one layer of the formulations P50% and P60%, respectively. The decrease in the opacity with the coating application can be justified by the coverage of the pores and voids of the paper, which in turn reduces the light scattering on the fibers matrix [41]. As already observed in the SEM images of the coated samples, the application of a second layer led to an increased coating coverage of the fibers, promoting a further reduction of the opacity values for both formulations (88.84 ± 1.14% and 75.1 ± 5.65% for P50% and P60%, respectively).

### 3.3. Air Barrier Properties

One of the main disadvantages of paper for packaging applications is its low air barrier properties, which can be improved by applying a coating [42]. Therefore, the air barrier properties of the coated papers prepared in this work and pristine base paper (for comparative purposes) were evaluated by measuring their Gurley’s air resistivity (Figure 5). The base paper showed an air resistivity of 7.74 ± 0.27 s, which is in line with the reported values for papers with similar grammage [43]. The application of one layer of the formulation P50% had a statistically significant impact (8.41 ± 0.19 s) on this parameter, but the application of the second layer (8.67 ± 0.24 s) did not have a substantial effect. The coating with one layer of the formulation P60% led to an increase of this parameter to 10.5 ± 0.64 s, confirming that the amount of applied coating indeed had an effect on the porosity of the paper, which impacted not only the opacity of the paper but also its air resistivity. When a second layer of this formulation was applied, the air resistivity was further increased (25.11 ± 3.52 s), indicating a significant reduction of the paper’s porosity for higher pick-up values, which is in line with the SEM analysis (Figure 3d) and the opacity reduction verified for these samples. It is expected that the increase in the number of layers would continue to progressively improve the air resistivity, up to a point when a continuous film would be obtained, covering the voids and pores of paper [44]. For instance, Sundar et al. [14] studied the application of PLA coatings to kraft paper, and an increase in the papers’ air resistivity from 62 s/100 mL to 110 s/100 mL was achieved by applying a coating with a pick-up of 6 g/m^2^. Although the values of air resistivity obtained by these authors were higher than the ones obtained in the present study, the improvement represents an increase of 77% in the paper’s air resistivity, while our best result (25.11 s for coated papers with two layers of P60%_2) denotes an increase of 224%.

### 3.4. Mechanical Properties

The good mechanical properties of paper are one of the reasons for its use as a packaging material; consequently, the application of a coating should not have a detrimental impact on those properties [44]. Figure 6 shows the burst index of the base and coated paper samples with both formulations. The application of formulation P50% did not have a statistically significant impact on the burst index of the coated papers (1.77 ± 0.14 kPa.m^2^/g), not even upon the application of a second layer (1.86 ± 0.01 kPa.m^2^/g). As regards formulation P60%, the application of one layer showed a statistically significant positive impact on the burst index of paper, increasing it from 1.88 ± 0.11 (base paper) to 2.1 ± 0.09 kPa.m^2^/g. However, the application of a second layer of the same formulation did not have a statistically significant impact (1.95 ± 0.05 kPa.m^2^/g). Nonetheless, the increase observed with the application of one layer of formulation P60% represents an improvement of 11.7%. This indicates that the application of this formulation not only maintains the mechanical performance of paper (burst index) but actually improves it, similar to the improvement reported when PLA was applied as a paper coating material [45], with the advantage that no volatile organic solvent is required for its application. Moreover, both P50% (which did not have a statistically significant impact on the burst index of the paper) and P60% formulations represent an advantage when compared with the use of commercially available PHB since its application as a paper coating was reported to have a negative impact in the burst index of paper by ≈30% [11].

### 3.5. Hydrophobic Properties

The major disadvantage of using paper as a packaging material is its intrinsic hydrophilicity [44]. So, to evaluate the effect of the application of this coating on the phobic behavior of the coated samples, water contact angle measurements and water absorption (using the Cobb test) were carried out. Figure 7 shows the results obtained for these two assays for the base paper and papers coated with the different formulations. The application of one layer of both formulations had a significant impact on the paper’s water contact angle, increasing the contact angle from 70.89 ± 4.56° (base paper) to 127.18 ± 2.9° and 125.94 ± 2.32° for P50% and P60%, respectively.

Interestingly, the application of a second layer led to contact angle values very similar to those obtained with a single layer (127.18 ± 2.9° and 125.94 ± 2.32° for P50% and P60%, respectively). This proves that, in this case, the deposition of one single layer is enough to fabricate papers with contact angles higher than 125°, which are values that indicate the achievement of hydrophobic properties. These results are in line with those obtained by Song et al. [46], which performed grafting polymerization of acrylic monomers (viz. butyl acrylate and 2-ethylhexyl acrylate) on cellulose fibers, achieving fibers with water contact angles of 130°, with a 4 wt% of grafted copolymer. However, in this case, the authors used plasma treatment on paper to induce the polymerization, which increases the cost and complexity of the process of both curing and deposition, as the authors needed to soak the paper for at least 1 h in the monomers to be grafted.

Despite the similar water contact angles obtained for the papers coated with P50% and P60%, significant differences were found in what regards the water barrier properties of the coated papers obtained with the two formulations and with a different number of layers (Figure 7b). The application of one layer of the formulation P50% reduced the water absorption from 108.22 ± 1.94 g/m^2^ (base paper) to 56.57 ± 7.69 g/m^2^, and the deposition of a second layer of the same formulation led to a further decrease to 18.76 ± 1.24 g/m^2^, which is in the same range of the values obtained for the papers coated with one layer of the formulation P60% (15.32 ± 0.74 g/m^2^). In fact, these two conditions, P50% with two layers and P60% with one layer, have similar pick-up values, viz. 10.39 and 14.17 g/m^2^, respectively. The paper coated with two layers of formulation P60% (pick-up of 31.85 g/m^2^) is the one that presents the lowest water absorption value, namely 10.8 g/m^2^. These results clearly indicate a clear relationship between the water barrier properties of these coated papers and the corresponding pick-up values. Thus, when the pick-up of the coated papers increases, it is verified a decrease in the water absorption values. Since the formulations with a higher amount of copolymer yielded coated papers with higher pick-ups, the adjustment of the amount of copolymer in the formulation can be used to increase the water absorption properties of paper.

Table 2 presents some of the most recent works regarding the application of paper coating formulations to improve paper’s hydrophobicity. Our formulations yielded papers with similar water barrier performance and hydrophobicity when compared with one of the best-reported coating formulations, namely that based on chitosan-*g*-poly(dimethylsiloxane) [47]. Despite the fact that our methodology requires a higher pick-up to achieve it, the use of a reactive solvent is a key advantage when compared to the need of organic solvents (e.g., dichloromethane and chloroform) [14,48] or even water [47,49]. So, the alternative investigated in this work highlights the use of UV irradiation and photopolymerizable formulations, which avoids the use of volatile organic solvents, their removal (high energy consumption), and water waste generation.

## 4. Conclusions

A solventless photopolymerizable formulation capable of imparting paper with high hydrophobicity and water and gas barrier properties has been developed using 2-ethylhexyl acrylate and isobornyl methacrylate.

The addition of the copolymer of the same monomers in the formulation allowed a better control of the pick-up of the coated papers, with the formulation having a higher copolymer content (60%) yielding coatings with higher pick-ups (up to 31.85 g/m^2^). Overall, the formulations had a minimal detrimental impact on the optical properties of paper with the increase in the coating pick-up. On the contrary, it was observed that the use of these formulations positively impacts the air barrier properties of paper where, e.g., two layers of the P60% formulation allowed to attain Gurley’s air resistance values of 25.11 s, which are significantly higher than that of base paper (7.74 s). The mechanical properties of the coated paper were not negatively affected by any of the formulations, indicating that the mechanical performance of the paper was maintained after the coating application and photopolymerization. The coated papers also showed excellent hydrophobic properties (water contact angle for all formulations > 125 °), noting that the use of a second coating layer of the formulation does not have a statistically significant impact on this parameter. The coated paper also presented excellent water barrier properties, with the application of one layer of the formulation P60% reducing the water absorption values from 108 g/m^2^ to 15 g/m^2^ and even more with the application of a second layer (10 g/m^2^).

In brief, the potentialities of a photopolymerizable coating based on the monomers 2-ethylhexyl acrylate and isobornyl methacrylate and a copolymer of these same monomers were demonstrated for the first time, showing that they are capable of yielding papers with high hydrophobicity and water barrier properties, without the usage of a traditional solvent (water or volatile organic solvents). Therefore, this strategy represents a step forward toward the development of new and greener methodologies for paper coating applications.

## Figures and Tables

**Figure 1 polymers-15-01069-f001:**
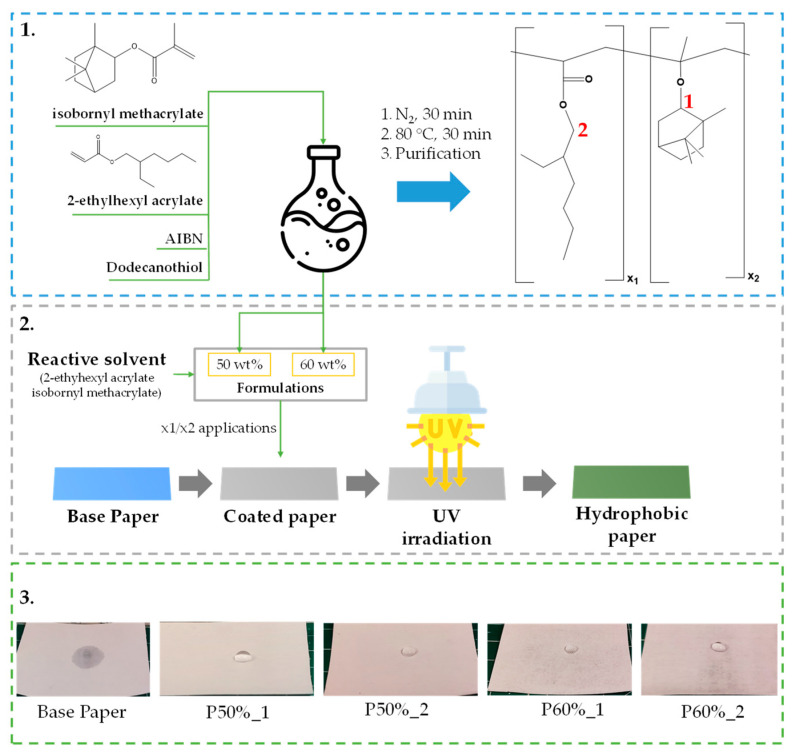
Scheme illustrating the work procedure: 1—copolymer synthesis and purification; 2—preparation and application of the formulations, followed by UV irradiation; 3—photos of the obtained coated papers highlighting their hydrophobicity. Numbers 1 and 2 in red in the structure of the copolymer represent the hydrogens that were used to calculate the copolymer composition by ^1^H NMR.

**Figure 2 polymers-15-01069-f002:**
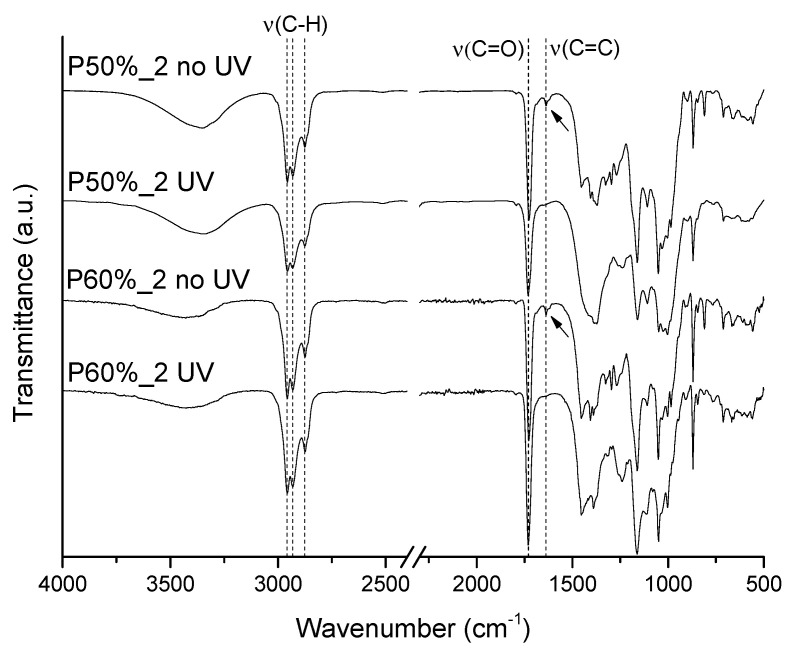
Normalized FTIR spectra (vibrational modes: ν = stretching) of the coated papers with different formulations (P50% and P60%) before and after UV irradiation.

**Figure 3 polymers-15-01069-f003:**
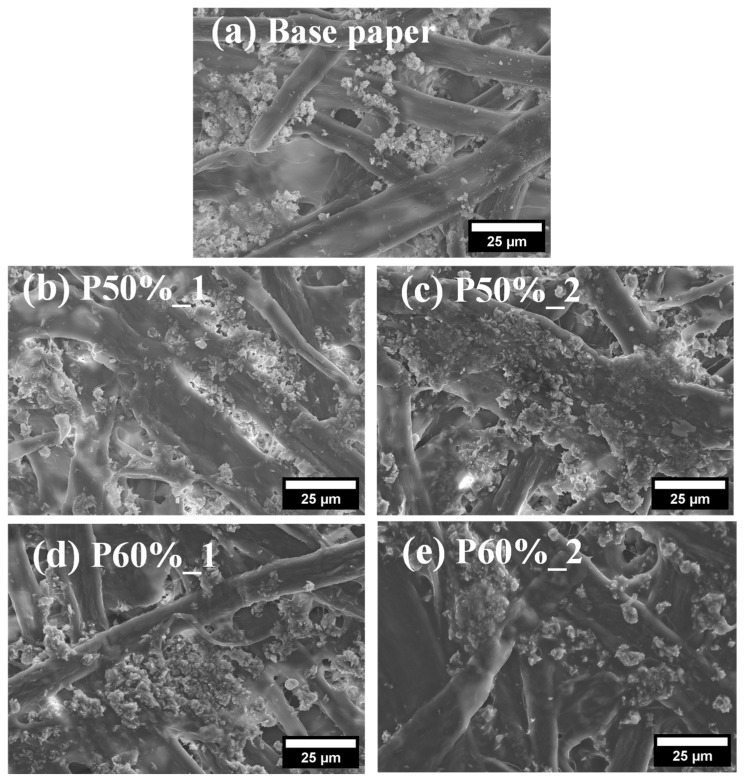
Surface SEM micrographs of base paper (**a**) and coated papers with one (**b**) and two (**c**) layers of the formulation P50%, and one (**d**) and two (**e**) layers of formulation P60%, at a magnification of 1000×.

**Figure 4 polymers-15-01069-f004:**
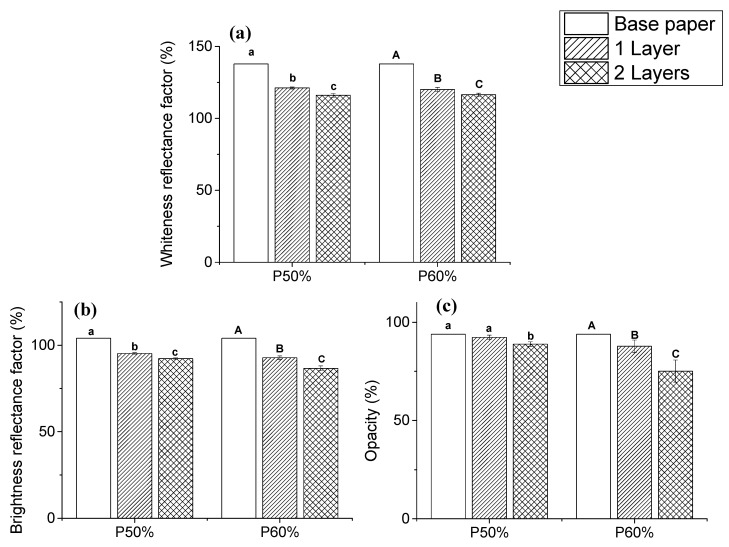
Optical properties of the base paper (blank) and coated papers with different formulations (50% and 60% of copolymer) and number of layers (1 layer—single line, 2 layers—double line): whiteness (**a**); brightness (**b**); and opacity (**c**). The results are expressed as mean ± SD of measurements. Statistical analysis was performed using a one-way ANOVA and the Tukey test, with a level of significance of *p* < 0.05, analyzed using the Origin Pro 9.0 software. Statistical analysis was performed independently for each formulation. The differences between samples with the same letter are not statistically significant.

**Figure 5 polymers-15-01069-f005:**
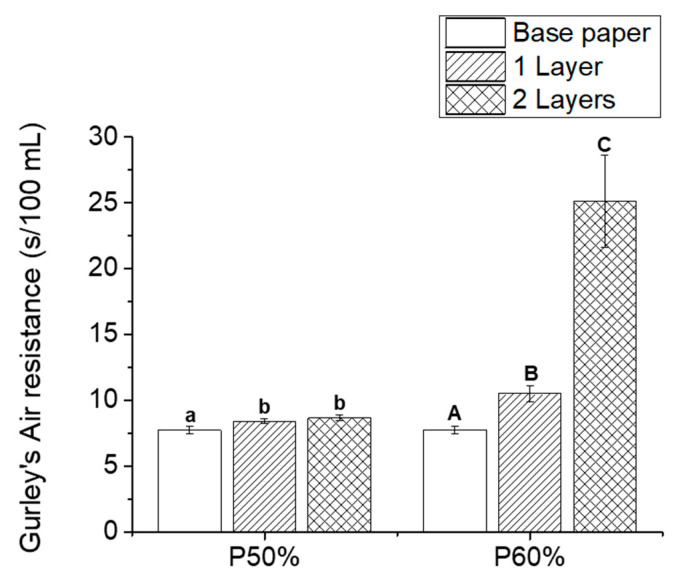
Gurley’s air resistivity results for base paper (blank) and coated samples with different formulations (50% and 60% of copolymer) and number of layers (1 layer—single line, 2 layers—double line). The results are expressed as mean ± SD of measurements. Statistical analysis was performed using a one-way ANOVA, and the Tukey test, with a level of significance of *p* < 0.05, analyzed using the Origin Pro 9.0 software. Statistical analysis was performed independently for each formulation. The differences between samples with the same letter are not statistically significant.

**Figure 6 polymers-15-01069-f006:**
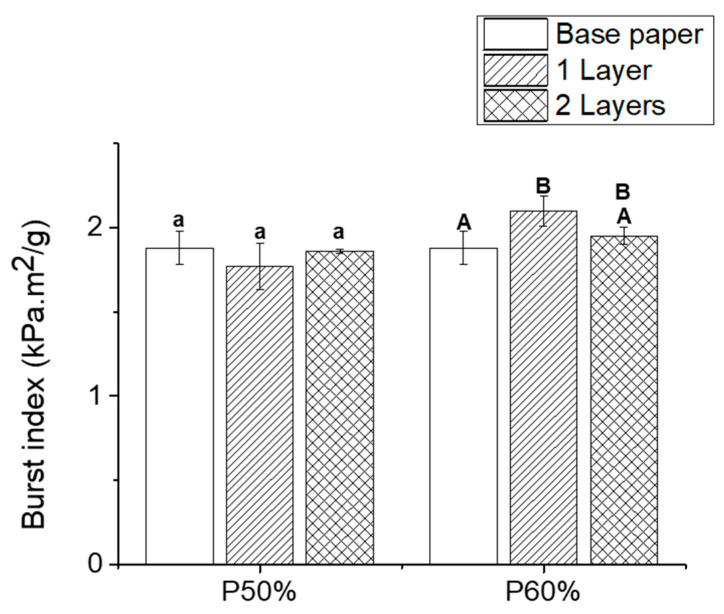
Burst index of base paper (blank) and coated paper samples with different formulations and number of layers (1 layer—single line, 2 layers—double line). The results are expressed as mean ± SD of measurements. Statistical analysis was performed using a one-way ANOVA, and the Tukey test, with a level of significance of *p* < 0.05, analyzed using the Origin Pro 9.0 software. Statistical analysis was performed independently for each formulation. The differences between samples with the same letter are not statistically significant.

**Figure 7 polymers-15-01069-f007:**
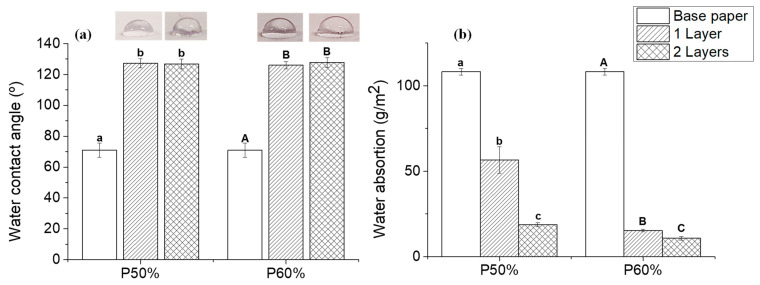
Hydrophobic properties of base paper (blank) and coated paper samples with different formulations and number of layers (1 layer—single line, 2 layers—double line). Mean contact angle with the corresponding digital photographs (**a**) and water absorption results obtained through Cobb test (**b**). The results are expressed as mean ± SD of measurements. Statistical analysis was performed using a one-way ANOVA and the Tukey test, with a level of significance of *p* < 0.05, analyzed using the Origin Pro 9.0 software. Statistical analysis was performed independently for each formulation. The differences between samples with the same letter are not statistically significant.

**Table 1 polymers-15-01069-t001:** Grammage and pick-up values of the different coated papers.

Sample Designation	Formulation	Number of Layers	Grammage (g/m^2^)	Pick-Up (g/m^2^)
Base paper	-	-	79.26 ± 0.26	-
P50%_1	P50%	1	85.98 ± 0.86	6.71 ± 0.34
P50%_2		2	89.74 ± 0.42	10.39 ± 0.59
P60%_1	P60%	1	93.24 ± 0.92	14.17 ± 0.92
P60%_2		2	110.99 ± 2.27	31.85 ± 2.47

**Table 2 polymers-15-01069-t002:** Examples of some of the most recent coating formulations applied to improve paper hydrophobicity.

Coating Material	Solvent	Pick-Up (g/m^2^)	Water Absorption (g/m^2^)	Contact Angle (°)	Ref.
Poly(2-ethylhexyl acrylate-*co*-isobornyl methacrylate)	2-ethylhexyl acrylate + isobornyl methacrylate	14.17	15.3	125	our work
Chitosan-*g*-poly(dimethylsiloxane)	Water	9.4	15.8	122.8	[47]
Poly(styrene-***co***-maleimide)	Water	10.6	34	100	[49]
Polylactic acid	Dichloromethane	6.0	10	75	[14]
Poly(-3-hydroxybutyrate-*co*-3-hydroxyvalerate)	Chloroform	8.5	-	85	[48]
Polycaprolactone	Chloroform	8.3	-	85	[48]

## Data Availability

The data presented in this study are available on request from the corresponding author.

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
