# Peer review of "Solventless Photopolymerizable Paper Coating Formulation for Packaging Applications"

_polymers, 2023, doi:10.3390/polym15051069_

Round 1

Reviewer 1 Report

·     -  Please confirm whether the ideas in introduction are smooth and the transition is natural. The reasons for the use of various raw materials are clearly explained, but the story is not good enough.

- The authors should re-organize the introduction to increase the readability and highlight the innovation of the manuscript compared with other works. Overall, the comparison with other studies are weak. Some other literature may help you (https://link.springer.com/article/10.1007/s10924-022-02740-7; https://www.sciencedirect.com/science/article/pii/S0926669023000109 )

-Why the authors didn't attach the 1H NMR spectra to this manuscript?

-Why OTR and WVP of the samples have not been measured? These factors are so important for packaging paper.

Author Response

  1. Please confirm whether the ideas in introduction are smooth and the transition is natural. The reasons for the use of various raw materials are clearly explained, but the story is not good enough.

REPLY: We would like to acknowledge the referee for the valuable comments and suggestions to improve our manuscript. As recommended, we rechecked the flow of the article introduction and made some changes to clarify the story.

  1. The authors should re-organize the introduction to increase the readability and highlight the innovation of the manuscript compared with other works. Overall, the comparison with other studies are weak. Some other literature may help you (https://link.springer.com/article/10.1007/s10924-022-02740-7; https://www.sciencedirect.com/science/article/pii/S0926669023000109)

REPLY: As mentioned above, we reformulate the introduction to make the reading clearer and highlight the novelty of the manuscript. However, the suggested articles were not included since they do not focus on photopolymerizable coatings for paper applications. Nonetheless, additional references were added to enhance the overall quality of the discussion.

  1. Why didn’t the authors attach the 1H NMR spectra to this manuscript?

REPLY: The 1H NMR spectrum was added as support information and referenced in the main document as Figure S1.

  1. Why OTR and WVP of the samples have not been measured? These factors are so important for packaging paper.

REPLAY: We understand that assessing the OTR and WVP in materials for packaging is very important. However, in this study, we couldn’t obtain access to the testing apparatus to execute them. Therefore, to have some idea regarding the impact of the coating on paper’s air barrier properties, we performed a Gurley test. The results are presented in Figure 5.

Reviewer 2 Report

In this manuscript, aiming the requirements of packaging application, authors developed a solventless photopolymerizable paper coating technique, which shows good potential for the fabrication of hydrophobic papers in packaging. In general, it is an interesting work and the manuscript is well organized. However, there are still some issues to be addressed. A moderate revision is required before its acceptance.

1.     There are many abbreviations. Please make sure the full name is provided when first appearance.

2.     When generally give the background of packaging, some recent and important articles should be included to supporting the statements: Biobased materials for food packaging; Nanomaterials 10 (1), 150, 2020; Packaging and degradability properties of polyvinyl alcohol/gelatin nanocomposite films filled water hyacinth cellulose nanocrystals; Nanomaterials 12 (18), 3158, 2022; etc.

3.     More descriptions should be provided for the sample preparations during the characterizations.

4.     Three-line table is suggested for a more scientific expression.

5.     The mechanism of the UV irradiation should be further clarified.

6.     It is better to insert the photo of the contact angle in Fig. 7.

7.     There are still some typos and grammar issues in the manuscript. Authors should carefully recheck the whole manuscript, such as “Error! Reference source not found.”, etc.

Author Response

  1. There are many abbreviations. Please make sure the full name is provided when first appearance.

REPLY: All abbreviations were checked, and full spelling was provided the first time they were mentioned.

  1. When generally give the background of packaging, some recent and important articles should be included to supporting the statements: Biobased materials for food packaging; Nanomaterials 10 (1), 150, 2020; Packaging and degradability properties of polyvinyl alcohol/gelatin nanocomposite films filled water hyacinth cellulose nanocrystals; Nanomaterials 12 (18), 3158, 2022; etc.

REPLY: We added a citation for the article, with the title “Biobased materials for food packaging”. However, we do not think the other suggestions were suitable to be cited in this work.

  1. More descriptions should be provided for the sample preparations during the characterizations.

REPLY: We do not understand this comment because all the details required for the replication of the experimental procedure, including the characterization section, were previously included in the manuscript. Moreover, most of the characterization procedures are well-known approaches for researchers that work in this field.

  1. Three-line table is suggested for a more scientific expression.

REPLY: We understand the reviewer comment. However, we use the table template provided by Polymers journal. Nevertheless, we changed the format of the tables to a three-line table, as requested.

  1. The mechanism of the UV irradiation should be further clarified.

Reply: Considering the type of monomers used, the mechanism involved is a typical textbook free radical polymerization mechanism which the readership of Polymers is familiar with. Thus, this information is considered redundant.

  1. It is better to insert the photo of the contact angle in Fig. 7.

REPLY: Digital photos of water droplets on the coated paper were added to figure 7 for a better representation of the contact angle values.

  1. There are still some typos and grammar issues in the manuscript. Authors should carefully recheck the whole manuscript, such as “Error! Reference source not found.”, etc.

REPLY: All typos, grammar issues, and the cross references of figures and tables in the text were checked.

Round 2

Reviewer 2 Report

 Accept in present form